# Coronavirus Disease 2019 (COVID-19): A Modeling Study of Factors Driving Variation in Case Fatality Rate by Country

**DOI:** 10.3390/ijerph17218189

**Published:** 2020-11-05

**Authors:** Jennifer Pan, Joseph Marie St. Pierre, Trevor A. Pickering, Natalie L. Demirjian, Brandon K.K. Fields, Bhushan Desai, Ali Gholamrezanezhad

**Affiliations:** 1Keck School of Medicine, University of Southern California, Los Angeles, CA 90033, USA; panjenni@usc.edu (J.P.); jstpierr@usc.edu (J.M.S.P.); tpickeri@usc.edu (T.A.P.); ndemirji@usc.edu (N.L.D.); bkfields@usc.edu (B.K.K.F.); bhushan.desai@med.usc.edu (B.D.); 2Department of Preventive Medicine, Keck School of Medicine, University of Southern California, Los Angeles, CA 90033, USA; 3Department of Integrative Anatomical Sciences, Keck School of Medicine, University of Southern California, Los Angeles, CA 90033, USA; 4Department of Radiology, Keck School of Medicine, University of Southern California, Los Angeles, CA 90033, USA

**Keywords:** COVID-19, SARS-CoV-2, pneumonia, computed tomography, case fatality rate, social distancing, smoking

## Abstract

*Background:* The novel Severe Acute Respiratory Syndrome Coronavirus-2 has led to a global pandemic in which case fatality rate (CFR) has varied from country to country. This study aims to identify factors that may explain the variation in CFR across countries. *Methods:* We identified 24 potential risk factors affecting CFR. For all countries with over 5000 reported COVID-19 cases, we used country-specific datasets from the WHO, the OECD, and the United Nations to quantify each of these factors. We examined univariable relationships of each variable with CFR, as well as correlations among predictors and potential interaction terms. Our final multivariable negative binomial model included univariable predictors of significance and all significant interaction terms. *Results:* Across the 39 countries under consideration, our model shows COVID-19 case fatality rate was best predicted by time to implementation of social distancing measures, hospital beds per 1000 individuals, percent population over 70 years, CT scanners per 1 million individuals, and (in countries with high population density) smoking prevalence. *Conclusion:* Our model predicted an increased CFR for countries that waited over 14 days to implement social distancing interventions after the 100th reported case. Smoking prevalence and percentage population over the age of 70 years were also associated with higher CFR. Hospital beds per 1000 and CT scanners per million were identified as possible protective factors associated with decreased CFR.

## 1. Introduction

On 31 December 2019, a pneumonia of increasing incidence and unknown etiology in Wuhan, China was reported to the World Health Organization (WHO). Investigations led to the discovery of a novel coronavirus, later dubbed Severe Acute Respiratory Syndrome Coronavirus-2 (SARS-CoV-2), which causes the pathology known as Coronavirus Disease 2019 (COVID-19) [1]. Despite initial containment measures recommended by the WHO in early January 2020, COVID-19 spread rapidly to other countries in the following weeks and was eventually classified as a global pandemic on 11 March 2020 [2,3]. Since then, it has posed major challenges to healthcare systems in affected countries around the world while crippling the global economy. At the time of our analysis, epidemiological data indicates that COVID-19 had since spread to 205 countries with over 3.1 million reported cases and over 227,000 deaths worldwide [4].

As effective antiviral therapies and vaccines remain unavailable, current efforts to halt the transmission of COVID-19 rely on social distancing, individual quarantine and isolation, and community containment measures [5]. Yet there has been great variation worldwide in the implementation of such measures. As demonstrated in China, the spread of the disease was slowed by effectively combining the largest quarantine ever implemented—over 200,000 people were either tracked via contact tracing or received medical observation as of 30 January 2020—with stringent community facemask use, limitations on social gatherings, isolation of affected workplace institutions, and lockdown of multiple public transportation outlets to isolate communities and towns with outbreaks [6,7]. However, this approach is resource-intensive and is less likely to be emulated by more liberal democracies [8]. Therefore, it may be useful to examine the differences in implementation of quarantine policies in different countries and their impact on disease mortality.

Numerous studies have examined the daily and cumulative number of confirmed cases by country and have analyzed variation in case-fatality rate (CFR), defined as number of deaths relative to number of confirmed cases. Estimates of CFR are placed at 2.3% in China, while estimates of infection-fatality ratio (IFR), which attempt to account for proportions of mild and asymptomatic disease, sits markedly lower at 0.1–0.94% [7,9,10,11]. However, because the exact proportions of mild and asymptomatic cases are variable and reliant on estimations in lieu of confirmatory testing, IFR-based metrics have yet to produce a reliable model [10,11]. Therefore, CFR remains a more concrete measure of describing and identifying predictive factors associated with disease mortality. CFR varies from country to country and, while multiple studies have found links to possible underlying factors driving changes in case mortality, a complete explanation of this variation remains unclear. Using country-specific data from global organizations, our study aims to identify factors that best explain differences in CFR among 39 highly impacted countries during the first five months of the COVID-19 pandemic. To the best of our knowledge, this is the first study to investigate multiple factors affecting CFR using country-specific data to drive our modeling.

## 2. Materials and Methods

### 2.1. Data Collection

We began by conducting a targeted literature search for potential risk factors for COVID-19 mortality through Pubmed, using variations on terms including “COVID-19 mortality risk factors” and “pandemic mortality risk factors”. Our approach was similar to a study conducted by Morales et al., investigating H1N1 influenza risk factors that varied by country [12]. In our review of studies on the current pandemic as well as past SARS and influenza pandemics, we identified 24 risk factors (Appendix A) for COVID-19 fatality that were worth investigating, including quarantine policies, air travel activity, age distribution, comorbidities, healthcare access, availability of diagnostic tests, cumulative and daily testing data, and environmental factors such as air pollution and climate [5,6,7,13,14,15,16,17,18,19,20,21,22]. Regarding quarantine policies, government responses varied by country. For purposes of this study, we defined this intervention as the first date per international news sources when recommendations were made or legislation was passed limiting gathering size, closing non-essential business, or encouraging social distancing (Appendix A). School closures and international travel bans were not considered as they only applied to certain individuals within each country’s population.

To calculate CFR, we used the total confirmed cases and deaths for a given country from Our World in Data [4]. As higher testing rates per capita could be associated with an increased record of mild cases, we included total cumulative tests and tests per 1000 to explore the relationship between CFR and testing capacity per capita. Additionally, we included socioeconomic factors such as GDP, level of education, and scientific production. To examine healthcare-related factors, we included hospital beds, physicians per 1000, and per capita healthcare expenditure. We also looked at the availability of CT scanners and radiologists per one million population, as CT chest imaging represents a possibly limited resource that can increase the diagnostic accuracy. This was particularly true during the first wave of the pandemic, during which reverse transcription polymerase chain reaction (RT-PCR) kits were limited and imaging was relied upon by multiple countries for diagnosis of the disease [23]. Finally, given recent data linking the comorbidities of obesity and chronic lung disease to increased disease severity and poorer outcomes in cases of COVID-19 [16,19,24], we included prevalence of obesity, chronic obstructive pulmonary disease (COPD), and tobacco use, along with particulate matter as a measure of air pollution.

We decided to include in our analysis all countries with over 5000 COVID-19 deaths at the time of writing [4]. This cutoff was chosen to generate a set of at least 20 countries located in all hemispheres with diverse quarantine measures, GDPs, and geographic locations. 39 countries were found to meet this criterion and were included in the study. For each country, data was generated for the 24 variables from the following sources: Our World in Data project [4], World Bank database [25], OECD database [26], United Nations World Population Prospects [27], Global Health Data Exchange [28], and various international news sources (Appendix A).

### 2.2. Statistical Analysis

We began by examining the distributions of and correlations among all variables. We found that population density, GDP in 2017, scientific production, total tests, air travel, percentage illiterate, and air pollution were highly positively skewed. To reduce the influence of extreme observations, we transformed these variables on the log scale, except for population density for which a square root transformation was sufficient. We next examined the univariable relationships between independent variables and case fatality rates to find candidate variables for our final multivariable model. We found the negative binomial model for case fatality rate to be appropriate, as the overdispersion parameter α was statistically significant in each of these models. We noted the significance of each variable in these univariable models and selected variables for a preliminary multivariable model at *p* < 0.15. We additionally wanted to examine whether the effect of candidate predictors was moderated by population density and time to quarantine. We therefore examined models with each independent variable and (1) an interaction with population density and (2) an interaction with days from 100th case to quarantine (dichotomized as >14 days). Days from 100th case to quarantine was dichotomized as it showed a nonlinear relationship with CFR, with higher CFR rates in the highest quartile (>14 days). Our preliminary multivariable model was created by including all univariable candidate predictors and then adding all significant interaction terms. A hands-on guided approach was used to check for any anomalies that arose when adding or removing variables from the final model. Our preliminary final model contained all multivariable predictors significant at *p* < 0.05. Before arriving at our final model, each variable excluded in the univariable step was added back one-by-one to ensure they were still nonsignificant predictors of case fatality rate. All analyses were performed in Stata (v15.0).

## 3. Results

Case fatality rate varied widely by country, as low as 0.6% and as high as 17.7% (M = 5.4%, SD = 4.3%). Means and standard deviations of candidate predictors are shown in Table 1, along with rate ratios from the univariable negative binomial model for each individual predictor. We found that percent population >70 years old, general mortality per 1000 individuals, percentage of population illiterate, percentage of population with HIV, and air pollution were significantly associated with case fatality rate in these univariable models. In addition, tests per 1000 individuals, percentage of population obese, smokers, tobacco users, and with HIV significantly interacted with population density.

We also found several instances of collinearity. For example, we found high correlations between: smoking prevalence and tobacco use prevalence (r = 0.92, *p* < 0.001); air travel and GDP (r = 0.96, *p* < 0.001); and percentage of population >70 years old was correlated with several variables such as general mortality (r = 0.80, *p* < 0.001), prevalence of COPD (r = 0.71, *p* < 0.001), life expectancy (r = 0.74, p < 0.001), physicians per capita (r = 0.73, *p* < 0.001), tests per capita (r = 0.88, *p* < 0.001), and GDP (r = 0.71, *p* < 0.001). When collinear variables were included in the model, we sequentially added them in one-by-one and evaluated the model fit; the variable producing the best fit was retained in the model.

Our final model included time from 100th case to quarantine (dichotomized >14 days), hospital beds per 1000 individuals, percentage population over 70 years, CT scanners per 1 million individuals (log-transformed), and interaction between smoking prevalence and population density. This model had good agreement between observed and predicted CFR values (Figure 1). We found that countries waiting over 14 days from the 100th case to quarantine had 1.5 times the case fatality rate of those that did not wait as long (*p* = 0.045), and each percentage increase in the population over 70 years was associated with 1.15-time increase in the case fatality rate (*p* < 0.001). Though proportion of population over 70 years was correlated with a slew of health-related variables, there were some that were predictive of case fatality rate above and beyond the proportion of the population that is elderly. We found that each additional hospital bed per 1000 individuals reduced the case fatality rate by 15% (RR = 0.85, *p* < 0.001), and that a 1-unit increase in the log number of CT scanners per million was associated with half the case fatality rate (RR = 0.49, *p* < 0.001). The deleterious effect of smoking on case fatality rate was significant, but only in countries with higher density (*p*-interaction < 0.001). To aid in interpretation, we calculated the rate ratio at the mean, and 0.5 SD below and above the mean, of square root transformed population density. These results are presented in Table 2 (Model 1).

We performed additional sensitivity analyses to explore the effect of (1) date of country being impacted by COVID-19 and (2) missing covariate data. First, we created a new model by including date of 100th case to examine any change in coefficients (Table 2, Model 2). It was suspected that CFR may be lower in countries that reached their 100th case later, as they may not have had sufficient time for the virus to act on individuals. We examined the correlation between date of 100th case and days-to-quarantine and found a negative relationship (r = −0.47, *p* = 0.003) after excluding China, which was impacted early but also had quick quarantine implementation (Figure 2). We also note that 13 countries were missing data on CT scanners in our model. To determine possible impacts of this missing data, we performed multiple imputation with 25 data sets using all complete variables in the data that were not included in our final models. Our final model changed slightly but not appreciably (Table 2, Model 3). Date of 100th case was not significant when added to this model (*p* = 0.66; model not shown).

## 4. Discussion

Our analysis of 24 variables relative to COVID-19 mortality across 39 countries suggests that the case fatality rate is related to a variety of country-specific factors, including time to implement social distancing measures after the 100th case, hospital beds per 1000 individuals, percentage population over 70 years, CT scanners per 1 million individuals, and smoking prevalence with high population density.

Social distancing interventions, such as increased case isolation and community contact reduction, have been shown to be highly effective in slowing the spread of the virus [29]. According to our model, countries that waited over 14 days to implement social distancing interventions after their 100th reported case saw an increased CFR (RR = 1.54, *p* = 0.045), consistent across all population densities. As COVID-19 spread has been shown to occur during the asymptomatic incubation period [30,31], the promptness of local and national government response in implementing quarantine policies may have played a crucial role in limiting human-to-human transmission. As respiratory failure from Acute Respiratory Distress Syndrome appears to be the leading cause of mortality [32], surges in severe COVID-19 cases have the potential to overwhelm the capacity of a country’s healthcare system to provide mechanical ventilation and other intensive resources [33,34]. Thus, timely implementation of social distancing measures may, in many cases, have delayed epidemic peak in regard to CFR by reducing exponential growth of cases [29]. However, we did see this effect attenuated when including date of 100th case in the model. While it may be possible that countries that were affected more recently may not have had time to fully experience the true extent of COVID-related deaths, we did find that countries affected by COVID-19 later were quicker to implement quarantine measures. Because of this, we cannot truly disentangle the effect of time-to-quarantine from date of 100th case.

With regards to comorbidities, our model predicts that a 10% increase in smoking prevalence more than doubles the CFR in countries with high population density (RR = 2.53, *p* < 0.001). There is a growing body of evidence associating increased risk of both increased severity of disease, ICU admission, and death among infected patients with smoking history, particularly active smokers [35,36]. Yet, it is unclear why our model correlated smoking prevalence with increased CFR strongly in high-density populations only. One possible explanation is that high density areas are more likely to suffer outbreaks sufficient to overwhelm the local hospital and ICU capacity, triggering mass casualty protocols with possible triaging of resource allocation in favor of patients with more favorable prognostic indicators [37,38]. Smoking history is often associated with other medical comorbidities, notably cardiovascular disease, which may contribute to a poorer overall prognostic presentation and thus less priority in a resource-poor scenario [36,39]. Finally, the effect that smoking may have as a potential risk factor is likely to be more exaggerated in densely populated regions where these vulnerable individuals interact with others in closer proximity and with higher frequency. Smoking prevalence may also be related to other factors that are impacted by population density. It will be necessary to further explore the effects of smoking and its sequalae on disease course to determine the additional considerations that should be given to patients with a history of smoking, particularly in areas of high population density.

Case fatality rate was reduced by 15% (RR = 0.85, *p* < 0.001) for each additional hospital bed per 1000 individuals, a reflection of resources available for delivering inpatient medical services. Studies on past influenza pandemics have shown that scarcity of healthcare resources and clinical infrastructure, particularly in rural or developing areas, is a major limitation to pandemic preparedness [40,41,42]. It is important to note that the capacity of a healthcare system is tied not only to infrastructure but also to the availability of providers; in order to increase capacity, it will be necessary to build a larger healthcare workforce to support more hospital beds and higher patient volume [41]. CT scanners represent another limited resource that appears to be a protective factor. Our model shows that a 1-unit increase in the log number of CT scanners per million was associated with half the case fatality rate (RR = 0.49, *p* < 0.001). A possible explanation for this protective effect is that CT scans have led to earlier detection of the disease, as early reports describe characteristic imaging features that are helpful in aiding diagnosis [43]. This may have been particularly advantageous in developing countries, where the capacity to develop and mass-distribute testing was limited during the first months of the pandemic [43,44,45], as well as for frontline providers in any country irrespective of testing ability as a means of providing earlier diagnosis [46,47]. Countries with more radiologists and CT scanners per capita are also likely to have increased availability of other health resources, and thus the variables may serve as proxy for other factors that reflect the robustness of a nation’s healthcare system.

Our model also suggests a positive relationship between CFR and percentage of the population over age 70, with every added percent increasing the CFR by a factor of 1.15. This variable is highly associated with general mortality (r = 0.80, *p* < 0.001), life expectancy (r = 0.74, *p* < 0.001), and COPD (r = 0.71, *p* < 0.001) at the country level and could potentially be viewed as a proxy for indicating a country with older, more vulnerable population. Observations on age and increased case mortality are consistent with multiple retrospective studies identifying advanced age as a potential risk factor for more severe disease and worsened prognosis [48,49]. There are multiple possible explanations for this observation. In addition to frailty and increased risk of having multiple co-morbidities, some studies suggest age-related declines in T and B cell function alongside preserved innate immunity may be contributory, with the resultant cytokine 2-dominant response triggering a pro-inflammatory state which increases mortality [50,51].

### Limitations and Future Directions

This modeling study used a cross-sectional, ecological dataset taken during the pandemic’s first wave. As such, most of our model’s limitations stem from the weaknesses of this approach. As with most data gathered in the first several months of the pandemic, accuracy was limited by the information available at that time and capacity to report cases in a timely manner may vary from country to country. This limitation affects estimates of CFR, which has been shown to require an assessment of the delay between infection and the reporting of case data as well as the extent to which death-related cases are underreported. Furthermore, CFR itself is best modeled as a continuous variable which changes over time as the disease spreads to areas which vary in risk due to population density and demographic, as well as how healthcare systems and government policies adapt to disease burden [52]. Finally, given the ecological nature of our study, it should be noted that our findings and discussion of risk factors do not necessarily reflect a relationship between these variables and probability of survival at the individual level.

In addition, our study was limited by the availability of datasets used for country-specific data. Some of the datasets used to generate data for the 24 variables did not include all countries. For example, although our univariate analysis did not find a significant correlation between CFR and total radiologists or radiologists per one million, our data on radiologists per country was limited to one study from 2008 that included only 26 of the 39 countries analyzed [53]. More data is needed on this subject, as early reports on the global response to COVID-19 have shown that computerized tomography (CT) of the chest may serve an integral role in the timely diagnosis of the disease, as well as in severity staging and monitoring of clinical course [43,44,47,54,55]. Another limitation arises from utilizing data pertaining to entire countries for risk factors that may vary within each country on a geographically smaller scale-among cities, for example. This is especially true for large countries such as the United States and China with wide variations in population density, resource availability, and environmental characteristics based on region. For the timing of isolation and quarantine measures, our study relied on multiple secondary news sources for specific dates (Appendix A), which, despite a standardized and systematic approach, is inherently less reliable than documentation from a single primary source.

Future studies are also needed to evaluate the effect and timing of government interventions on disease spread and CFR. While our results suggest a possible relationship between early government implementation of social distancing measures and reduced CFR, it remains unclear whether differences in the type or stringency of these measures appreciably influences CFR. Timing of quarantine measures is also important, and a comparison of specific state-imposed measures as well as their timing relative to the date of first confirmed case and other milestones of cumulative case growth could represent a strong follow-up to our findings. We also suggest a closer look at the relationship between smoking and its association with severe disease, as well as the extent to which such risk factors affect comprehensive care under triage protocols in resource-restricted circumstances.

## 5. Conclusions

Using country-based multivariate modeling, our study found significant correlations between increased CFR and smoking prevalence, percentage of a population over the age of 70 years, increased time to implementation of social distancing or stay-at-home measures, as well as decreased CFR and hospital beds per 1000 and CT scanners per million. Notably, CFR appears to increase significantly for every day after the 100th documented case where governmental precautions are not put in place. More research is needed on the relationship between the timing and effectiveness of government precautions and case fatality rate, as well as the role that hospital bed capacity and CT scanner availability play in reducing case fatality. The relationship between population density and smoking prevalence, as well as the influence smoking has on disease course, is also worth further exploration.

## Figures and Tables

**Figure 1 ijerph-17-08189-f001:**
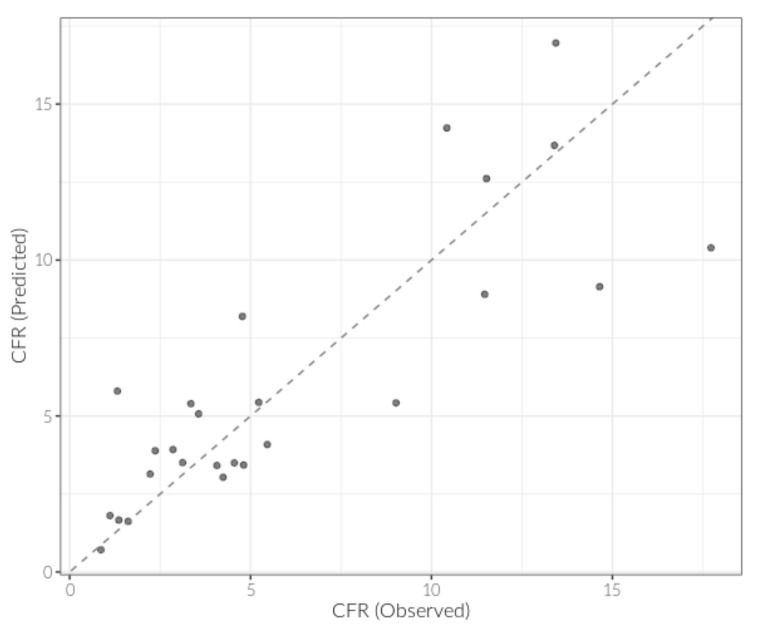
Final model (Model 1) of predicted values plotted against observed values of case fatality rate. These two variables were correlated at r = 0.84.

**Figure 2 ijerph-17-08189-f002:**
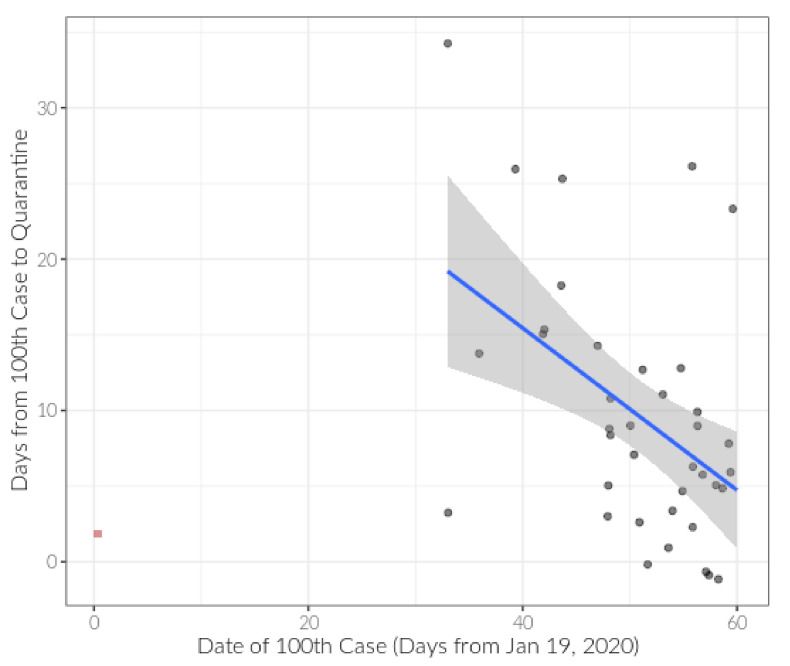
Days from 100th case to quarantine plotted against date of 100th case (in days from 19 January 2020; China’s 100th case). The two variables are correlated (r = −0.47, *p* = 0.003) after removing China (red square).

**Table 1 ijerph-17-08189-t001:** Descriptive statistics (mean & SD) for candidate predictor variables, rate ratios (SE) for univariable relationships between predictors and case fatality rate, and *p*-value for the interaction of each predictor variable with population density (square root transformed). For comparability, all rate ratios reflect the effect of the standardized predictor on case fatality rate.

Variable	Mean (SD)	Rate Ratio (SE)	*p* (Density Interaction)
Percent population >70 years old ^†^	8.9 (4.7)	1.33 (0.16) *	-
Population density	158.7 (148.2)	1.14 (0.32)	-
Population size ^†^	135.4 × 10^6^ (310 × 10^6^)	1.07 (0.14)	-
GDP in 2017 ($) ^†^	1.82 × 10^12^ (3.57 × 10^12^)	1.23 (0.16)	-
GDP per capita in 2017	29761 (22379)	1.11 (0.15)	-
Healthcare expenditure per capita	2849 (2735)	1.17 (0.16)	-
Scientific production ^†^	53393 (91189)	1.20 (0.15)	-
Hospital beds per 1000	3.95 (2.91)	0.92 (0.14)	-
Physicians per 1000	2.78 (1.26)	1.16 (0.14)	-
General mortality per 1000	7.82 (2.62)	1.44 (0.21) *	-
Life expectancy	78.7 (4.3)	1.21 (0.14)	-
CT scanners per 1 million	26.6 (22.2)	0.75 (0.13)	-
Radiologists ^†^	5863 (14180)	1.20 (0.20)	-
Radiologists per 1 million	64.1 (43.2)	1.25 (0.20)	-
Total tests ^†^	330013 (325817)	1.15 (0.14)	-
Tests per 1000	12.0 (9.4)	1.04 (0.15)	0.04
Median age	36.3 (6.8)	1.23 (0.14)	-
Days from 100th case to quarantine	9.5 (8.4)	1.26 (0.18)	-
Air travel ^†^	93587 (165381)	1.05 (0.13)	-
Education	73.5 (19.2)	0.88 (0.14)	-
Percent Illiterate ^†^	4.5 (8.1)	0.75 (0.09) *	-
Percent Obese	21.1 (8.5)	0.99 (0.15)	0.005
Percent Smokers	20.3 (6.2)	1.08 (0.14)	0.03
Percent Tobacco Users	23.3 (8.0)	1.11 (0.15)	0.06
Percent HIV	0.2 (0.3)	1.30 (0.18) *	0.001
Percent COPD	5.4 (2.3)	1.23 (0.15)	-
Air pollution ^†^	27.2 (34.0)	0.68 (0.09) **	-

^†^ Log-transformed variable was used for Rate Ratio, * *p* < 0.05, ** *p* < 0.01.

**Table 2 ijerph-17-08189-t002:** Final multivariable negative binomial model predicting case fatality rate. Rate ratios and 95% confidence intervals are presented. Smoking prevalence is evaluated at the mean of (square root transformed) population density, 0.5 SD below (low, approximately 65 per km^2^), and 0.5 SD above (high, approximately 200 per km^2^). Model I contains our final estimates without imputation (*n* = 26), Model II additionally adjusts for date of 100th case, and Model III shows the results from our final model on imputed CT scanner data (*n* = 39).

Variable	Model I	Model II	Model III
	RR (95% CI)	RR (95% CI)	RR (95% CI)
Prevalence smoking (10% population increase)			
at low population density	1.00 (0.69, 1.44)	1.13 (0.80, 1.61)	0.96 (0.69, 1.33)
at mean population density	1.59 (0.99, 2.56)	1.72 (1.12, 2.65)	1.33 (0.90, 1.96)
at high population density	2.53 (1.32, 4.87)	2.62 (1.46, 4.70)	1.83 (1.09, 3.07)
>14 days from 100th case to quarantine	1.54 (1.01, 2.35)	1.23 (0.78, 1.92)	1.57 (1.01, 2.43)
Hospital beds per 1000 individuals	0.85 (0.78, 0.92)	0.84 (0.77, 0.90)	0.58 (0.45, 0.74)
Percent population >70 years	1.15 (1.08, 1.23)	1.12 (1.03, 1.20)	1.13 (1.07, 1.20)
CT scanners per 1 million individuals (log)	0.49 (0.34, 0.67)	0.44 (0.32, 0.60)	0.67 (0.46, 0.98)
Date of 100th case (days)	-	0.96 (0.92, 0.99)	-

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
