# Peer review of "Coronavirus Disease 2019 (COVID-19): A Modeling Study of Factors Driving Variation in Case Fatality Rate by Country"

_ijerph, 2020, doi:10.3390/ijerph17218189_

Round 1

Reviewer 1 Report

The purpose of this study is to examine the potential risk factor for the case fatality rate of SARS-COV-2. This includes a well-written structure and a timely manner topic. However, there are several issues that might need to revise this paper.

The major concern for this study is how authors have identified the 24 risk factors. Although the introduction well explained the current situation and necessity of CFR research, I couldn't find any rationale for how authors selected these variables. The authors should provide any rationale or references for each independent variable in the introduction or the methods.

Moreover, there are some concerns regarding the methods and results. 

- How authors control the multicollinearity?

- What is the rationale for selected variables (p<0.15)? Is this variable selection is from all independent variables in table 1? If so, what is your final model? 

- Table 2, Prevalence of smoking at different population density is not clear. How and why the authors used this separation? Is this stratification also applicable to other variables?

- The authors used population density, but how you identify it? Since it is county level, the population density for the entire county might bring a lot of geographical issues.

- Authur used the Case Fatality "Rate" as a dependant variable, but it is not integers. Why the author used negative binomial regression instead of linear regression?

- I am not sure whether figure 2 is meaningful or necessary. I agree that 100th cases can be represented as the speed of the virus spread, but it is also influenced by the number of population.  Moreover, each country has a different starting date of 1st case, so it might be an issue. 

Author Response

Response to Reviewer’s Comments

Reviewer 1

The purpose of this study is to examine the potential risk factor for the case fatality rate of SARS-COV-2. This includes a well-written structure and a timely manner topic. However, there are several issues that might need to revise this paper.

The major concern for this study is how authors have identified the 24 risk factors. Although the introduction well explained the current situation and necessity of CFR research, I couldn't find any rationale for how authors selected these variables. The authors should provide any rationale or references for each independent variable in the introduction or the methods.

                Thank you for your comments.

We addressed this concern by elaborating on our approach to the literature search that led to our identification of the 24 risk factors:

We began by conducting a targeted literature search for potential risk factors for COVID-19 mortality through Pubmed, using variations on terms including “COVID-19 mortality risk factors” and “pandemic mortality risk factors”. Our approach was similar to Morales et al.’s study on H1N1 influenza risk factors that varied by country. 

Moreover, there are some concerns regarding the methods and results. 

- How authors control the multicollinearity?

We excluded variables from consideration that had a high variance inflation factor (VIF), reassessing the VIF values after sequentially excluding the variable with the highest VIF. Because of the generally high correlations among all variables, we chose a conservative cutoff of 2.5 for this. Reference for 2.5 VIF: (Johnston R, Jones K, Manley D. Confounding and collinearity in regression analysis: a cautionary tale and an alternative procedure, illustrated by studies of British voting behaviour. Qual Quant. 2018;52(4):1957-1976. doi:10.1007/s11135-017-0584-6)

- What is the rationale for selected variables (p<0.15)? Is this variable selection is from all independent variables in table 1? If so, what is your final model? 

A relaxed p-value is commonly used in building a predictive regression model. (e.g., Jobson, J. Dave. Applied multivariate data analysis: regression and experimental design. Springer Science & Business Media, 2012.). I don’t understand the last question; our final model is listed in Table 2.

- Table 2, Prevalence of smoking at different population density is not clear. How and why the authors used this separation? Is this stratification also applicable to other variables?

On lines 114-116 we discuss how we went through and examined the interaction of each variable with population density and days to quarantine. We thus explored the interaction of each predictor with population density. Smoking was the only predictor that showed a significant interaction in our final model.

- The authors used population density, but how you identify it? Since it is county level, the population density for the entire county might bring a lot of geographical issues.

 Discussion of this limitation has been added as follows:

Another limitation arises from utilizing data pertaining to entire countries for risk factors that may vary within each country on a geographically smaller scale– among cities, for example. This may be especially true for large countries such as the United States and China with wide variations in population density, resource availability, and environmental characteristics based on region.

- Author used the Case Fatality "Rate" as a dependant variable, but it is not integers. Why the author used negative binomial regression instead of linear regression?

Negative binomial regression is preferable when modeling rate outcomes. Technically this model had # deaths as the outcome, with an offset for # of cases, which is definitionally a model for the CFR. 

- I am not sure whether figure 2 is meaningful or necessary. I agree that 100th cases can be represented as the speed of the virus spread, but it is also influenced by the number of population.  Moreover, each country has a different starting date of 1st case, so it might be an issue. 

The purpose of Figure 2 is to illustrate that time until quarantine is related to date of 100th case. 

Reviewer 3

The aim of this manuscript was to identify risk factors that can predict case fatality rates (CFRs) of COVID-19 in various countries, and to do that, the authors collected country-specific data on 24 potential risk factors affecting CFR from 39 countries with over 5000 reported COVID-19 cases at the time of data collection. A negative binomial regression model was built and included univariate predictors of significance and all significant interaction terms. The model revealed that smoking prevalence, percent of a population over the age of 70 years and time to implementation of social distancing or stay-at-home measures were associated with increased CFR, and hospital beds per 1000 and CT scanners per million were associated with decreased CFR. The authors concluded that timely implementation of social distancing measures is key to delay the epidemic peak in regards to CFR. Upon review of the manuscript, I find that the study was appropriately designed, the statistical analysis was carefully conducted, and the evidence presented generally support the conclusions. A few more clarifications and minor analyses would be helpful to solidify and confirm the authors’ findings.

                Thank you for your comments.

The authors used correlation coefficients (I assume they are Spearman’s correlation coefficients, and is the cutoff, needs to be clarified) to identify multicollinearity among predictors. Although correlation coefficient is widely used as an indicator of multicollinearity, it only measures bivariate linear relationships, and multicollinearity is multivariate. I suggest that, in addition to the correlation coefficients, the authors measure the variance inflation factor (VIF) to formally detect multicollinearity.

This relates to Reviewer 1’s comment on multicollinearity, copied below. We now use the VIF as a metric to exclude collinear variables:

We excluded variables from consideration that had a high variance inflation factor (VIF), reassessing the VIF values after sequentially excluding the variable with the highest VIF. Because of the generally high correlations among all variables, we chose a conservative cutoff of 2.5 for this. Reference for 2.5 VIF: (Johnston R, Jones K, Manley D. Confounding and collinearity in regression analysis: a cautionary tale and an alternative procedure, illustrated by studies of British voting behaviour. Qual Quant. 2018;52(4):1957-1976. doi:10.1007/s11135-017-0584-6)

Relating to comment 1, hospital beds per 1000 and CT scanner per million might be positively correlated (although the Spearman’s coefficient might be less than 0.7). It is observed that when including countries with missing information on CT scanner availability using multiple imputations in Model III, both the regression coefficients of hospital beds and CT scanners changed dramatically from Model I (effect size increased for hospital beds and decreased for CT scanners), which might suggest that the multicollinearity issue was still present.

 We examined the VIF for all variables in the final model and found the value to be 1.97 for hospital beds and 1.57 for CT scanners.

In Table 2 Model II, date of 100th case was significant (95% CI of estimated relative risk not covering 1), but the authors claimed in the Results section that “the inclusion of date of 100th case was not significant in the model”. It is unclear how the authors assessed the model fit and further clarification is warranted. Also regarding the goodness-of-fit of the final model, the authors used a hands-on approach, by adding/removing individual predictors and testing their p-values, to derive the final model. This strategy, although is not exactly the stepwise regression, might still result in a local optimum in terms of model fit given the number of potential predictors. Granted that in the high dimensional setting, we can never know which variables are truly predictive, some methods like Lasso with cross-validation can substantially reduce the risk of a misspecified model. If the variables that the authors selected in the final model also have non-zero coefficients in Lasso, it will further support the feature selection conducted by the authors.

Lasso regression showed that many of the variables in the final model had nonzero coefficients (e.g., percent population over 70, CT scanners, population density, smoking prevalence). However we did find that the results of the Lasso regression were highly unstable, and varied greatly depending on factors such as scaling of our variables (e.g., using a raw vs. z-score for the variable). Therefore, we do not wish to rely too heavily on the results of the Lasso regression.

Some of the country-specific environmental and sociological information, like healthcare resources, education and air pollution, varies in different regions especially for large countries like Brazil, China, Russia and USA. The authors should acknowledge this limitation in their discussions.

This relates to a comment Reviewer 1 made. Discussion of this limitation has been added as follows:

Another limitation arises from utilizing data pertaining to entire countries for risk factors that may vary within each country on a geographically smaller scale– among cities, for example. This may be especially true for large countries such as the United States and China with wide variations in population density, resource availability, and environmental characteristics based on region.

Reviewer 2 Report

The manuscript prepared by Pan et al. describes a modeling study for COVID-19 in identifying key factors that influence the case fatality rate. The study analyzed 24 COVID-19-related factors across 39 countries and suggests that the case fatality rate depends on a variety of country-specific factors, including the time to implement social distancing measures after the 100th case, and smoking prevalence to mention a few. 

The analysis is based on a wide series of factors and the authors have done a nice job in describing the results carefully and clearly. Given the status of the disease, it is important to have data that convey the importance of factors such as social distancing in preventing the spread of the disease. Maybe the authors could discuss or expand a little bit more on the wearing mask factors if they have concrete data...

As a very minor point, in the supplemental information, please add the first page with the title of the manuscript including authors and affiliations. Additionally, in the reference list, please revise "6. ,  World Population Prospects..." as "6.  World Population Prospects..." Check all the references as this typo is all over the list.

Author Response

(The authors gave the same response as above.)

Reviewer 3 Report

The aim of this manuscript was to identify risk factors that can predict case fatality rates (CFRs) of COVID-19 in various countries, and to do that, the authors collected country-specific data on 24 potential risk factors affecting CFR from 39 countries with over 5000 reported COVID-19 cases at the time of data collection. A negative binomial regression model was built and included univariate predictors of significance and all significant interaction terms. The model revealed that smoking prevalence, percent of a population over the age of 70 years and time to implementation of social distancing or stay-at-home measures were associated with increased CFR, and hospital beds per 1000 and CT scanners per million were associated with decreased CFR. The authors concluded that timely implementation of social distancing measures is key to delay the epidemic peak in regards to CFR. Upon review of the manuscript, I find that the study was appropriately designed, the statistical analysis was carefully conducted, and the evidence presented generally support the conclusions. A few more clarifications and minor analyses would be helpful to solidify and confirm the authors’ findings.

Comments:

  1. The authors used correlation coefficients (I assume they are Spearman’s correlation coefficients, and is the cutoff, needs to be clarified) to identify multicollinearity among predictors. Although correlation coefficient is widely used as an indicator of multicollinearity, it only measures bivariate linear relationships, and multicollinearity is multivariate. I suggest that, in addition to the correlation coefficients, the authors measure the variance inflation factor (VIF) to formally detect multicollinearity.
  2. Relating to comment 1, hospital beds per 1000 and CT scanner per million might be positively correlated (although the Spearman’s coefficient might be less than 0.7). It is observed that when including countries with missing information on CT scanner availability using multiple imputations in Model III, both the regression coefficients of hospital beds and CT scanners changed dramatically from Model I (effect size increased for hospital beds and decreased for CT scanners), which might suggest that the multicollinearity issue was still present.
  3. In Table 2 Model II, date of 100th case was significant (95% CI of estimated relative risk not covering 1), but the authors claimed in the Results section that “the inclusion of date of 100th case was not significant in the model”. It is unclear how the authors assessed the model fit and further clarification is warranted.
  4. Also regarding the goodness-of-fit of the final model, the authors used a hands-on approach, by adding/removing individual predictors and testing their p-values, to derive the final model. This strategy, although is not exactly the stepwise regression, might still result in a local optimum in terms of model fit given the number of potential predictors. Granted that in the high dimensional setting, we can never know which variables are truly predictive, some methods like Lasso with cross-validation can substantially reduce the risk of a misspecified model. If the variables that the authors selected in the final model also have non-zero coefficients in Lasso, it will further support the feature selection conducted by the authors.
  5. Some of the country-specific environmental and sociological information, like healthcare resources, education and air pollution, varies in different regions especially for large countries like Brazil, China, Russia and USA. The authors should acknowledge this limitation in their discussions.

Author Response

(The authors gave the same response as above.)

Round 2

Reviewer 1 Report

I think the authors have tried their best to address the reviewer's comments. Obviously, the fundamental problem highlighted by other reviewers as well remains (i.e. variable selection, ecological fallacy etc.) but I feel that is now an editorial decision.